# Artificial Intelligence for Autonomous Molecular Design: A Perspective

**DOI:** 10.3390/molecules26226761

**Published:** 2021-11-09

**Authors:** Rajendra P. Joshi, Neeraj Kumar

**Affiliations:** Computational Biology Group, Biological Science Division, Pacific Northwest National Laboratory, 902 Battelle Blvd, Richland, WA 99352, USA; rajendra.joshi@pnnl.gov

**Keywords:** autonomous workflow, therapeutic design, computer aided drug discovery, computational modeling and simulations, quantum mechanics and quantum computing, artificial intelligence, machine learning, deep learning, machine reasoning and causal inference and causal reasoning

## Abstract

Domain-aware artificial intelligence has been increasingly adopted in recent years to expedite molecular design in various applications, including drug design and discovery. Recent advances in areas such as physics-informed machine learning and reasoning, software engineering, high-end hardware development, and computing infrastructures are providing opportunities to build scalable and explainable AI molecular discovery systems. This could improve a design hypothesis through feedback analysis, data integration that can provide a basis for the introduction of end-to-end automation for compound discovery and optimization, and enable more intelligent searches of chemical space. Several state-of-the-art ML architectures are predominantly and independently used for predicting the properties of small molecules, their high throughput synthesis, and screening, iteratively identifying and optimizing lead therapeutic candidates. However, such deep learning and ML approaches also raise considerable conceptual, technical, scalability, and end-to-end error quantification challenges, as well as skepticism about the current AI hype to build automated tools. To this end, synergistically and intelligently using these individual components along with robust quantum physics-based molecular representation and data generation tools in a closed-loop holds enormous promise for accelerated therapeutic design to critically analyze the opportunities and challenges for their more widespread application. This article aims to identify the most recent technology and breakthrough achieved by each of the components and discusses how such autonomous AI and ML workflows can be integrated to radically accelerate the protein target or disease model-based probe design that can be iteratively validated experimentally. Taken together, this could significantly reduce the timeline for end-to-end therapeutic discovery and optimization upon the arrival of any novel zoonotic transmission event. Our article serves as a guide for medicinal, computational chemistry and biology, analytical chemistry, and the ML community to practice autonomous molecular design in precision medicine and drug discovery.

## 1. Introduction

Synthesizing and characterizing small molecules in a laboratory with desired properties is a time-consuming task [1]. Until recently, experimental laboratories have been mostly human operated; they relied completely on the experts of the field to design experiments, carry out characterization, analyze, validate, and conduct decision making for the final product. Moreover, the experimental process involves a series of steps, each requiring several correlated parameters that need to be tuned [2,3], which is a daunting task, as each parameter set conventionally demands individual experiments. This has slowed down the discovery of high-impact small molecules and/or materials, in some case by decades, with possible implications for diverse fields, such as in energy storage, electronics, catalysis, drug discovery, etc.

Moreover, the high-impact materials of today come from exploring only a fraction of the known chemical space. Larger portions of the chemical space are still uncovered, and it is expected to contain exotic materials with the potential to bring unprecedented advances to state-of-the-art technologies. Exploring such a large space with conventional experiments will take time and a lot of resources [4,5,6,7]. In this scenario, complete automation of laboratories is long overdue and has been used with limited success in the past [8,9,10,11,12]. The concept of laboratory automation is not new [13]. It was used with limited success for material discovery in the past. More recently, automation has re-emerged as the approach of potential interest due to the significant development in computing architecture, sophisticated material synthesis, and characterization techniques, increasing the successful adoption of deep learning based models in physical and biological science domains. Automating the computational design of small molecules that integrates physics-based simulations and optimization with ML approaches is a feasible and efficient alternative instead; it significantly contributes in expediting autonomous molecular design.

High throughput quantum mechanical calculations, such as density functional theory (DFT), based simulations are the first step towards this goal of providing insight into larger chemical space and have shown some promise in accelerating novel molecule discovery. However, the physics based modeling still requires human intelligence for different decision-making processes, and for instance, it cannot autonomously guide small-molecule therapeutic design steps, thus slowing down the entire process. In addition, the inverse design of molecules is equally difficult with quantum mechanical simulations alone. The amount of data produced by these high throughput methods is so large that it cannot be analyzed in real-time with conventional methods. Autonomous computational design and characterization of molecules is more important in the scenarios where existing experimental/computational approaches are inefficient [14,15].

One such particular example is the challenge associated with identifying new metabolites in a biological sample from mass spectrometry data, which requires mapping the fragmented spectra of novel molecules to the existing spectral library, making it slow and tedious. In many cases, such references libraries do not exist, and an ML-integrated, automated workflow could be an ideal choice to deploy for the rapid identification of metabolites and the expansion of the existing libraries for future reference. Such a workflow has shown the early ability to quickly screen molecules and accurately predict their properties for different applications. The synergistic use of high throughput methods in a closed loop with machine-learning-based methods capable of inverse design is considered vital for autonomous and accelerated discovery of molecules [11].

In this contribution, we discuss how computational workflows for autonomous molecular design can guide the bigger goal of laboratory automation through active learning approaches. At first, we assess the performance of current state-of-the-art artificial intelligence (AI)-guided molecular design tools, mainly focusing on small molecule for therapeutic design and discovery. We start with an extensive discussion of popular molecular representation with various formulation and data generation tools used in advanced ML and deep learning (DL) models. We also benchmark the physics informed predictive ML by comparing various property predictions, which is critical for small-molecule design. In the end, we highlighted the cutting edge AI tools to utilize these ML models for inverse design with desired properties.

## 2. Results and Highlights

### 2.1. Components of Computational Autonomous Molecular Design Workflow

The workflow for computational autonomous molecular design (CAMD) must be an integrated and closed-loop system (Figure 1) with: (i) efficient data generation and extraction tools, (ii) robust data representation techniques, (iii) physics-informed predictive machine learning models, and (iv) tools to generate new molecules using the knowledge learned from steps i–iii. Ideally, an autonomous computational workflow for molecule discovery would learn from its own experience and adjust its functionality as the chemical environment or the targeted functionality changes through active learning. This can be achieved when all the components work in collaboration with each other, providing feedback while improving model performance as we move from one step to other.

For data generation in CAMD, high-throughput density functional theory (DFT) [16,17] is a common choice mainly because of its reasonable accuracy and efficiency [18,19]. In DFT, we typically feed in 3D structures to predict the properties of interest. Data generated from DFT simulations is processed to extract the more relevant structural and properties data, which are then either used as input to learn the representation [20,21] or as a target required for the ML models [22,23,24]. Data generated can be used in two different ways: to predict the properties of new molecules using a direct supervised ML approach and to generate new molecules with the desired properties of interest using inverse design. CAMD can be tied with supplementary components, such as databases, to store the data and visualize it. The AI-assisted CAMD workflow presented here is the first step in developing automated workflows for molecular design. Such an automated pipeline will not only accelerate the hit identification and lead optimization for the desired therapeutic candidates but can actively be used for machine reasoning to develop transparent and interpretable ML models. These workflows, in principle, can be combined intelligently with experimental setups for computer-aided synthesis or screening planning that includes synthesis and characterization tools, which are expensive to explore in the desired chemical space. Instead, experimental measurements and characterization should be performed intelligently for only the AI-designed lead compounds obtained from CAMD.

The data generated from inverse design in principle should be validated by using an integrated DFT method for the desired properties or by high throughput docking with a target protein to find out its affinity in the closed-loop system, then accordingly update the rest of the CAMD. These steps are then repeated in a closed loop, thus improving and optimizing the data representation, property prediction, and new data generation component. Once we have confidence in our workflow to generate valid new molecules, the validation step with DFT can be bypassed or replaced with an ML predictive tool to make the workflow computationally more efficient. In the following, we briefly discuss the main component of the CAMD, while reviewing the recent breakthroughs achieved.

### 2.2. Data Generation and Molecular Representation

ML models are data-centric—the more data, the better the model performance. A lack of accurate, ethically sourced well-curated data is the major bottleneck limiting their use in many domains of physical and biological science. For some sub-domains, a limited amount of data exists that comes mainly from physics-based simulations in databases [25,26] or from experimental databases, such as NIST [27]. For other fields, such as for bio-chemical reactions [28], we have databases with the free energy of reactions, but they are obtained with empirical methods, which are not considered ideal as ground truth for machine learning models. For many domains, accurate and curated data does not exist. In these scenarios, slightly unconventional yet very effective approaches of creating data from published scientific literature and patents for ML have recently gained adoption [29,30,31,32]. These approaches are based on the natural language processing (NLP) to extract chemistry and biology data from open sources published literature. Developing a cutting edge NLP-based tool to extract, learn, and reason the extracted data would definitely reduce timeline for high throughput experimental design in the lab. This would significantly expedite the decision making based on the existing literature to set up future experiments in a semi-automated way. The resulting tools based on human–machine teaming is much needed for scientific discovery.

### 2.3. Molecular Representation in Automated Pipelines

Robust representation of molecules is required for accurate functioning of the ML models [33]. An ideal molecular representation should be unique, invariant with respect to different symmetry operations, invertible, efficient to obtain, and capture the physics, stereo chemistry, and structural motif. Some of these can be achieved by using the physical, chemical, and structural properties [34], which, all together, are rarely well documented so obtaining this information is considered cumbersome task. Over time, this has been tackled by using several alternative approaches that work well for specific problems [35,36,37,38,39,40] as shown in Figure 2. However, developing universal representations of molecules for diverse ML problems is still a challenging task, and any gold standard method that works consistently for all kind of problems is yet to be discovered. Molecular representations primarily used in the literature falls into two broad categories: (a) 1D and/or 2D representations designed by experts using domain specific knowledge, including properties from the simulation and experiments, and (b) iteratively learned molecular representations directly from the 3D nuclear coordinates/properties within ML frameworks.

Expert-engineered molecular representations have been extensively used for predictive modeling in the last decade, which includes properties of the molecules [41,42], structured text sequences [43,44,45] (SMILES, InChI), molecular fingerprints [46], among others. Such representations are carefully selected for each specific problem using domain expertise, a lot of resources, and time. The SMILES representation of molecules is the main workhorse as a starting point for both representation learning as well as for generating expert-engineered molecular descriptors. For the latter, SMILES strings can be used directly as one hot encoded vector to calculate fingerprints or to calculate the range of empirical properties using different open source platforms, such as RDkit [47] or chemaxon [48], thereby bypassing expensive features generation from quantum chemistry/experiments by providing a faster speed and diverse properties, including 3D coordinates, for molecular representations. Moreover, SMILES can be easily converted into 2D graphs, which is the preferred choice to date for generative modeling, where molecules are treated as graphs with nodes and edges. Although significant progress has been made in molecular generative modeling using mainly SMILES strings [43], they often lead to the generation of syntactically invalid molecules and are synthetically unexplored. In addition, SMILES are also known to violate fundamental physics and chemistry-based constraints [49,50]. Case-specific solutions to circumvent some of these problems exist, but a universal solution is still unknown. The extension of SMILES was attempted by more robustly encoding rings and branches of molecules to find more concrete representations with high semantical and syntactical validity using canonical SMILES [51,52], InChI [44,45], SMARTS [53], DeepSMILES [54], DESMILES [55], etc. More recently, Kren et al. proposed 100% syntactically correct and robust string-based representation of molecules known as SELFIES [49], which has been increasingly adopted for predictive and generative modeling [56].

Recently, molecular representations that can be iteratively learned directly from molecules have been increasingly adopted, mainly for predictive molecular modeling, achieving chemical accuracy for a range of properties [34,57,58]. Such representations as shown in Figure 3 are more robust and outperform expert-designed representations in drug design and discovery [59]. For representation learning, different variants of graph neural networks are a popular choice [37,60]. It starts with generating the atom (node) and bond (edge) features for all the atoms and bonds within a molecule, which are iteratively updated using graph traversal algorithms, taking into account the chemical environment information to learn a robust molecular representation. The starting atom and bond features of the molecule may just be one hot encoded vector to only include atom-type, bond-type, or a list of properties of the atom and bonds derived from SMILES strings. Yang et al. achieved the chemical accuracy for predicting a number of properties with their ML models by combining the atom and bond features of molecules with global state features before being updated during the iterative process [61].

Molecules are 3D multiconformational entities, and hence, it is natural to assume that they can be well represented by the nuclear coordinates as is the case of physics-based molecular simulations [62]. However, with coordinates, the representation of molecules is non-invariant, non-invertible, and non-unique in nature [35] and hence not commonly used in conventional machine learning. In addition, the coordinates by itself do not carry information about the key attribute of molecules, such as bond types, symmetry, spin states, charge, etc., in a molecule. Approaches/architectures have been proposed to create robust, unique, and invariant representations from nuclear coordinates using atom-centered Gaussian functions, tensor field networks, and, more robustly, by using representation learning techniques [34,58,63,64,65,66], as shown in Figure 3.

Chen et al. [34] achieved chemical accuracy for predicting a number of properties with their ML models by combining the atom and bond features of molecules with global state features of the molecules and are updated during the iterative process. The robust representation of molecules can also only be learned from the nuclear charge and coordinates of molecules, as demonstrated by Schutt et al. [58,63,65]. Different variants (see Equation (Equation 1)) of message passing neural networks for representation learning have been proposed, with the main differences being how the messages are passed between the nodes and edges and how they are updated during the iterative process using hidden states $h_{v}^{t}$. Hidden states at each node during the message passing phase are updated using
(1)$$m_{v}^{t+1}=\sum M_{t}\left(h_{v}^{t},h_{w}^{t},h_{vw}^{t}\right),\;\;\;h_{v}^{t+1}=S_{t}\left(h_{v}^{t},m_{v}^{t+1}\right)$$
where $M_{t}$ and $S_{t}$ are the message and vertex update functions, whereas $h_{v}^{t}$ and $h_{vw}^{t}$ are the node and edge features. The summation runs over all the neighbor of *v* in the entire molecular graph. This information is used by a readout phase to generate the feature vector for the molecule, which is then used for the property prediction.

These approaches, however, require a relatively large amount of data and computationally intensive DFT optimized ground state coordinates for the desired accuracy, thus limiting their use for domains/datasets lacking them. Moreover, representations learned from a particular 3D coordinate of a molecule fail to capture the conformer flexibility on its potential energy surface [66], thus requiring expensive multiple QM-based calculations for each conformer of the molecule. Some work in this direction based on semi-empirical DFT calculations to produce a database of conformers with 3D geometry has been recently published [66]. This, however, does not provide any significant improvement in predictive power. These methods, in practice, can be used with empirical coordinates generated from SMILES using RDkit/chemaxon but still require the corresponding ground state target properties for building a robust predictive modeling engine as well as optimizing the properties of new molecules with generative modeling.

Moreover, in these physics-based models, the cutoff distance is used to restrict the interaction among the atoms to the local environments only, hence generating local representations. In many molecular systems and for several applications, explicit non-local interactions are equally important [67]. Long-range interactions have been implemented in convolutional neural networks; however, they are known to be inefficient in information propagation. Matlock et al. [68] proposed a novel architecture to encode non-local features of molecules in terms of efficient local features in aromatic and conjugated systems using gated recurrent units. In their models, information is propagated back and forth in the molecules in the form of waves, making it possible to pass the information locally while simultaneously traveling the entire molecule in a single pass. With the unprecedented success of learned molecular representations for predictive modeling, they are also adopted with success for generative models [57,69].

### 2.4. Physics-Informed Machine Learning

Physics-informed machine learning (PIML) is the most widely studied area of applied mathematics in molecular modeling, drug discovery, and medicine [58,63,65,70,71,72,73,74,75,76]. Depending upon whether the ML architecture requires the pre-defined input representations as input features or can learn their own input representation by itself, PIML can be broadly classified into two sub-categories. The former is well covered in several recent review articles [70,71,72,73,74,75]. We will focus only on the latter, which has been increasingly adopted in predictive machine learning recently with unprecedented accuracy for a range of properties and datasets. A number of related approaches for predictive feature/property learning have been proposed in recent years under the umbrella term graph-based models so-called graph neural networks (GNNs) [77,78,79] and extensively tested on different quantum chemistry benchmark datasets. GNN for predictive molecular modeling consists of two phases: representation learning and property prediction, integrated end-to-end in a way to learn the meaningful representation of the molecules while simultaneously learning how to use the learned feature for the accurate prediction of properties. In the feature-learning phase, atoms and bond connectivity information read from the nuclear coordinates or graph inputs are updated by passing through a sequence of layers for robust chemical encoding, which are then used in subsequent property prediction blocks. The learned features can than be processed using dimensionality reduction techniques before using them in a subsequent property prediction block, as shown in Figure 4.

In one of the first works on embedded feature learning, Schütt et al. [63] used the concept of many body Hamiltonians to devise the size extensive, rotational, translational, and permutationally invariant deep tensorial neural network (DTNN) architecture for molecular feature learning and property prediction. Starting with the embedded atomic number and nuclear coordinates as input, and after a series of refinement steps to encode the chemical environment, their approach learns the atom-centered Gaussian-basis function as a feature that can be used to predict the atomic contribution for a given molecular property. The total property of the molecule is the sum over the atomic contribution. They demonstrated chemical accuracy of 1 kcal mol${}^{-1}$ in the total energy prediction for relatively small molecules in the QM7/QM9 dataset that contains only H, C, N, O, and F atoms.

Building on DTNN, Schütt et al. [58] also proposed a SchNet model, where the interactions between the atoms are encoded using a continuous filter convolution layer before being processed by filter generating neural networks. The predictive power of their model was further extended for electronic, optical, and thermodynamic properties of molecules in the QM9 dataset compared to only the total energy in DTNN, achieving state-of-the-art chemical accuracy in 8 out of 12 properties. The improved accuracy was observed over a related approach of Gilmer et al. [37], known as message passing neural network (MPNN), on a number of properties except polarizability and electronic spatial extent. In contrast to the SchNet/DTNN model, which learns atom-wise representation of the molecule, MPNN learns the global representation of molecules from the atomic number, nuclear coordinates, and other relevant bond-attributes and uses it for the molecular property prediction. It is critical to mention that MPNN is more accurate for the intensive properties (α, $\langle R^{2}\rangle$) where the decomposition into individual atomic contributions is not required. The performance of SchNet is further improved by Jørgensen et al. [80] by making edge features inclusive of the atom receiving the message.

In another related model, Chen et al. [34] proposed an integrated framework with unique feature update steps that work equally well for molecules and solids. They used several atom attributes and bond attributes and then combined it with the global state attribute to learn the feature representation of molecules. It was claimed that their method is outperforming the SchNet model in 11 out of 13 properties, including U0, U, H, and G in the benchmark QM9 dataset. However, they trained their model for respective atomization energies (P − $n_{X}X_{p}$, P = U0, U, H, and G) in contrast to the parent U0, U, H, and G trained model of Schnet. Based on our extensive assessment, a fair comparison of the model should be made between the similar quantities. These models also demonstrated that a model trained for predicting a single property of molecules with a graph-based model will always outperform the model optimized for predicting all the properties simultaneously. Other variants of MPNN are also published in the literature with slight improvements in accuracy for predicting some of the properties in the QM9 dataset over the parent MPNN [61,80]. The key features of a few benchmark models with their advantages and disadvantages are listed in Table 1. One particular approach is of Jorgenson et al. [80], where they extended the SchNet model in a way that the message exchanged between the atoms depends not only on the atom sending it but also on the atom receiving it. The comparison of mean absolute errors obtained from some of the benchmark models with their target chemical accuracy are reported in Table 2. This shows that the appropriate ML models, when used with the proper representation of molecules and a well-curated accurate dataset, a well-sought state-of-the-art chemical accuracy from machine learning can be achieved.

### 2.5. Inverse Molecular Design

To achieve the long overdue goal of exploring a large chemical space, accelerated molecular design, and generation of molecules with desired properties, inverse design is unavoidable. It is generally known that a molecule should have specific functionalities for it to be an effective therapeutic candidate against a particular disease, but in many cases, new molecules that host such functionalities are not easily known with a direct approach. Furthermore, the pool where such molecules may exist is astronomically large [81,82,83] (approx. 10${}^{60}$ molecules), making it impossible to explore each of them by quantum mechanics-based simulations or experiments.

In such scenarios, inverse design is of significant interest, where the focus is on quickly identifying novel molecules with desired properties in contrast to the conventional, so-called direct approach where known molecules are explored for different properties. In inverse design, we usually start with the initial dataset, for which we know the structure and properties, and map this to a probability distribution and then use it to generate new, previously unknown candidate molecules with desired properties very efficiently. Inverse design uses optimization and search algorithms [84,85] for the purpose and, by itself, can accelerate the lead molecule discovery process, which is the first step for any drug development. This paradigm holds even more promise when used in a closed loop with synthesis, characterization, and different test tools in such a way that each of these steps receives and transmits feedback concurrently, thus improving each other over time. This has shown some promise recently by substantially reducing the timeline for the commercialization of molecules from its discovery to days, which is otherwise known to span over a decade in most cases. In one recent work, Zhavoronkov et al. [1] designed, developed, and tested a workflow that integrates deep reinforcement learning with experimental synthesis, characterization, and test tools for the de novo design of drug molecules as potential inhibitors of the discoidin domain receptor-1 in 21 days. Such a paradigm shift in the design of drugs is possible only because of recently developed deep generative model architectures. Here, we briefly discuss some of the breakthrough architectures along with the recent applications in drug discovery.

Variational autoencoders [86] (VAEs) and its different variants have been extensively used for generating small molecules with optimal physio-chemical and biological properties. VAEs consist of an encoder and decoder network, where the encoder functions as a compression tool for compressing high-dimensional discrete molecular representations to a continuous vector in low-dimensional latent space, whereas the decoder recreates the original molecules from the compressed space. Within VAEs, recurrent neural networks (RNN) [87] and convolution neural networks (CNN) [88] are commonly used as encoding networks, whereas several RNN-based architectures, such as GRU and LSTM, are used as the decoder network. RNN independently has also been used to generate molecules. Bombarelli et al. [86] first used VAEs to generate molecules in the form of SMILES strings from latent space while simultaneously predicting their properties. For property prediction, they coupled the encoder–decoder network with the predictor network, which uses the vector from latent space as an input. SMILES strings generated from their VAEs do not always correspond to valid molecules. To improve on this, Kusner et al. [89] proposed a variant of VAEs known as the grammar VAE that imposes a constraint on SMILES generation by using context-free grammars rules. Both of these works employed string-based molecular representations. More recent works have focused on using molecular graphs as input and output for variational auto-encoders [90] using different variants of VAEs, among others [89,90,91], such as stacked auto-encoder, semi-supervised deep autoencoders, adversial autoencoder, and Junction Tree Variational Auto-Encoder (JT-VAE), for generating molecules for drug discovery. In JT-VAE [91], tree-like structures are generated from the valid sub-graph components of molecules and encoded along with a full graph to form two complementary latent spaces: one for the molecular graph and another for the corresponding junction tree. These two spaces are then used for hierarchical decoding, generating 100% valid small molecules. Further improvement on this includes using JT-VAE in combination with auto-regressive and graph-to-graph translation methods for valid large-molecule generation [92].

Generative adversarial networks (GANs) are another class of NN popular for generating molecules [93,94,95]. They consist of generative and discriminative models that work in coordination with each other where the generator is trained to generate a molecule and the discriminator is trained to check the accuracy of the generated molecules. Kadurin et al. [95] successfully first used the GAN architecture for de novo generation of molecules with anti-cancer properties, where they demonstrated higher flexibility, more efficient training, and processing of a larger dataset compared to VAEs. However, it uses unconventional binary chemical compound feature vectors and requires cumbersome validation of output fingerprints against the PubChem chemical library. Guimaraes et al. [96] and Sanchez-Lengeling et al. [97] used a sequence-based generative adversarial network in combination with reinforcement learning for molecule generation, where they bias the generator to produce molecules with desired properties. The works of Guimaraes et al. and Sanchez-Lengeling et al. suffer from several issues associated with a GAN, including mode collapse during training, among others. Some of these issues can be eliminated by using the reinforced adversarial neural computer method [98], which extends their work. Similar to VAEs, GANs have also been used for molecular graph generation, which is considered more robust compared to SMILES string generation. Cao et al. [94] non-sequentially and efficiently generated the molecular graph of small molecules with high validity and novelty from a jointly trained GAN and reinforcement learning architectures. Maziarka et al. [92] proposed a method for graph-to-graph translation, where they generated 100% valid molecules identical with the input molecules but with different desired properties. Their approach relies on the latent space trained for JT-VAE and a degree of similarity of the generated molecules to the starting ones can be tuned. Mendez-Lucio et al. [99] proposed conditional generative adversarial networks to generate molecules that produce a desired biological effect at a cellular level, thus bridging the system’s biology and molecular design. A deep convolution NN-based GAN [93] was used for de novo drug design targeting types of cannabinoid receptors.

Generative models, such as GANs, RNNs, and VAEs, have been used together with reward-driven and dynamic decision making reinforcement learning (RL) techniques in many cases with unprecedented success in generating molecules. Popova et al. [100] recently used deep-RL for the de novo design of molecules with desired hydrophobicity or inhibitory activity against Janus protein kinase 2. They trained a generative and a predictive model separately first and then trained both together using an RL approach by biasing the model for generating molecules with desired properties. In RL, an agent, which is a neural network, takes actions to maximize the desired outcome by exploring the chemical space and taking actions based on the reward, penalties, and policies setup to maximize the desired outcome. Olivecrona et al. [101] trained a policy-based RL model for generating the bioactives against dopamine receptor type 2 and generated molecules with more than 95% active molecules. Furthermore, taking an example of the drug Celecoxib, they demonstrated that RL can generate a structure similar to Celecoxib even when no Celecoxib was included in the training set. De novo drug design has so far only focused on generating structures that satisfy one of the several required criteria when used as a drug. Stahl et al. [102] proposed a fragment-based RL approach employing an actor-critic model for generating more than 90% valid molecules while optimizing multiple properties. Genetic algorithms (GAs) have also been used for generating molecules while optimizing their properties [103,104,105,106]. GA-based models suffer from stagnation while being trapped in at the regions of local optima [107]. One notable work alleviating these problems is by Nigam et al. [56], where they hybridize a GA and a deep neural network to generate diverse molecules while outperforming related models in optimization.

All of the generative models discussed above generate molecules in the form of 2D graphs or SMILES strings. Models to generate molecules directly in the form of 3D coordinates have also recently gained attention [57,108,109]. Such generated 3D coordinates can be directly used for further simulation using quantum mechanics or by using docking methods. One of such first models is proposed by Niklas et al. [57], where they generate the 3D coordinates of small molecules with light atoms (H, C, N, O, F). They then use the 3D coordinates of the molecules to learn the representation to map it to a space, which is then used to generate 3D coordinates of the novel molecules. Building on this for a drug discovery application, we recently proposed a model [69] to generate 3D coordinates of molecules while always preserving the desired scaffolds, as depicted in Figure 5. This approach has generated synthesizable drug-like molecules that show a high docking score against the target protein. Other scaffold-based models to generate molecules in the form of 2D graphs/SMILES strings are also published in the literature [110,111,112,113,114].

Recently, with the huge interest in the development of architecture and algorithms required for quantum computing, quantum version of generative models such as the quantum auto-encoder [115] and quantum GANs [116] have been proposed, which carry huge potential, among others, for drug discovery. The preliminary proof of concept work of Romero et al. [115,116] shows that it is possible to encode and decode molecular information using a quantum encoder, demonstrating generative modeling is possible with quantum VAEs, and more work, especially in the development of supporting hardware architecture, is required in this direction.

### 2.6. Protein Target Specific Molecular Design

The efficacy and potency of generated molecules against a target protein should be examined by predicting protein–ligand interactions (PLIs) and estimating key biophysical parameters. Figure 6 shows some of the computational methods frequently used in the literature (independently or together) for PLI prediction. Computationally, high throughput docking simulations [117,118,119] are most efficient and are used to numerically quantify and rank the interaction between the protein and ligand in terms of a docking score. These scores are based on the binding affinity of the ligand with the protein target and are used as the primary filter to narrow down high-impact candidates before performing more expensive simulations. Docking simulations are commonly used in combination with more accurate approaches to avoid false positives for pose prediction. Molecular mechanics (MM) simulations are another popular choice [120] but lack the accuracy that is generally required for making concrete decisions. Recently, all atoms molecular dynamics (MD) and hybrid QM/MM approach are increasingly adopted for studying protein–ligand interactions. It considers QM calculations for simulating the ligands and vicinity of protein where it docks while uses MM for simulating the rest of protein structure, providing improved accuracy over classical MM/docking simulations. Performing QM simulation even only for ligands and protein vicinity is computationally very expensive compared to relatively quick docking simulations. To expedite, QM simulations for ligands/protein vicinity can be replaced with state-of-art ML-based predictive model that has recently achieved chemical accuracy in predicting several properties of small molecules.

In this regards, several deep learning architectures have been used for efficient and accurate predictions of PLI parameters. These models vary among each other depending upon how protein or ligands are represented within the model [121,122,123,124]. For instance, Karimi et al. [125] proposed a semi-supervised deep learning model for predicting binding affinity by integrating RNN and CNN, wherein proteins are represented by an amino acid sequence and ligands in the form of SMILES strings. Other studies have used graph representations of ligand molecules with a string-based sequence representation of proteins [126,127]. Recently, Lim et al. [128] used a distance-aware GNN that incorporates 3D coordinates of both ligands and protein structures to study PLI outperforming existing models for pose prediction. The development and deployment of robust and accurate PLI models within a closed loop should be conducted in a way that encodes 3D coordinates of both protein and generated ligand molecules while simultaneously including and differentiating each ligand–residue interaction. This is important for accurately predicting the desired PLI interactions and biophysical parameters while designing high throughput novel molecules. It will contribute to efficiently narrow down the candidates during lead optimization, which ultimately will be subjected to further experimental characterization before it can be used for pre-clinical studies.

## 3. Conclusions and Future Perspectives

The success of current ML approaches depends on how accurately we can represent a chemical structure for a given model. Finding a robust, transferable, interpretable, and easy-to-obtain representation that obeys the physics and fundamental chemistry of the molecules that work for all different kinds of applications is a critical task. If such a spatial representation is available, it would save lot of resources while increasing the accuracy and flexibility of molecular representations. Efficiently using such representations with robust and reproducible ML architectures will provide a predictive modeling engine that would be ethically sourced with molecules metadata. Once a desired accuracy for diverse molecular systems for a given property prediction is achieved, it can routinely be used as an alternative to expensive QM-based simulations or experiments. In the chemical and biological sciences, a major bottleneck for deploying ML models is the lack of sufficiently curated data under similar conditions that is required for training the models. Finding architecture that works consistently well enough for a relatively small amount of data is equally important. Strategies such as active learning (AL) and transfer learning (TL) are ideal for such scenarios to tackle problems [129,130,131,132,133]. Graph-based methods for end-to-end feature learning and predictive modeling have been successfully used on small molecules consisting of lighter atoms. For larger molecules, robust representation learning and molecule generation parts must include non-local interactions, such as Van der Waals and H-bonding, while building predictive and generative models.

Equally important is developing and tying a robust, transferable, and scalable state-of-the-art platform for inverse molecular design in a closed loop with a predictive modeling engine to accelerate the therapeutic design, ultimately reducing the cost and time required for drug discovery. Many of the ML models used for inverse design use single biochemical activity as the criteria to measure the success of a generated candidate therapeutic, which is in contrast to a real clinical trial, where small-molecule therapeutics are optimized for several bio-activities simultaneously, leading to multi-objective optimization. Our contribution serves as inspiration to develop a CAMD workflow that should be engineered in a way to optimize multiple objective functions while generating and validating therapeutic molecules. Validation of all the newly generated lead molecules for a given target or disease-based models, if characterized by experiments or quantum mechanical simulations, is an very expensive task. We need to find ways to auto-validate molecules (using an inbuilt robust predictive model), which would be ideal to save resources and expedite molecular design. In addition, CAMD workflows should be able to quantify the uncertainty associated with it using statistical measures. For an ideal case, such uncertainty should decrease over the time as it learns from its own experience and reason in series of closed-loop experiments.

Currently, CAMD workflows are generally built and trained with a specific goal in mind. Such workflows need to be re-configured and re-trained to work for different objectives in therapeutic design and discovery. Designing and engineering a single automated CAMD setup for multiple experiments (multi-parameter optimization) through transfer learning is a challenging task, which can hopefully be improved based on the scalable computing infrastructure, algorithm, and more domain-specific knowledge. It would be particularly very helpful for the domains where a relatively small amount of data exist. Having such a CAMD infrastructure, algorithm and software stack would speedup end-to-end antiviral lead design and optimization for any future pandemics, such as COVID-19.

## Figures and Tables

**Figure 1 molecules-26-06761-f001:**
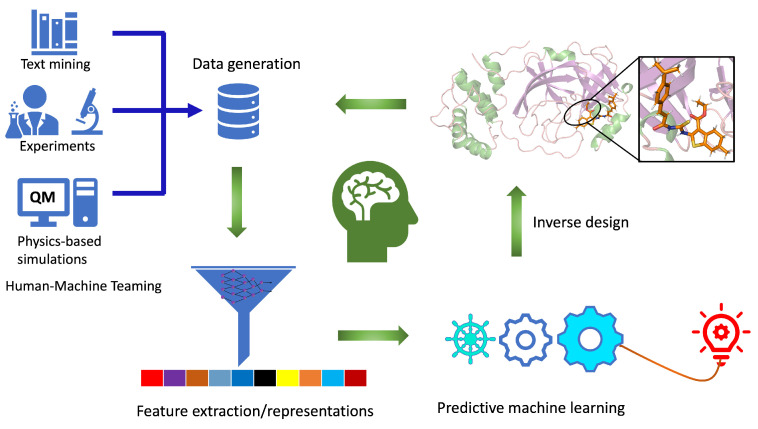
Closed-loop workflow for computational autonomous molecular design (CAMD) for medical therapeutics. Individual components of the workflow are labeled. It consists of data generation, feature extraction, predictive machine learning and an inverse molecular design engine.

**Figure 2 molecules-26-06761-f002:**
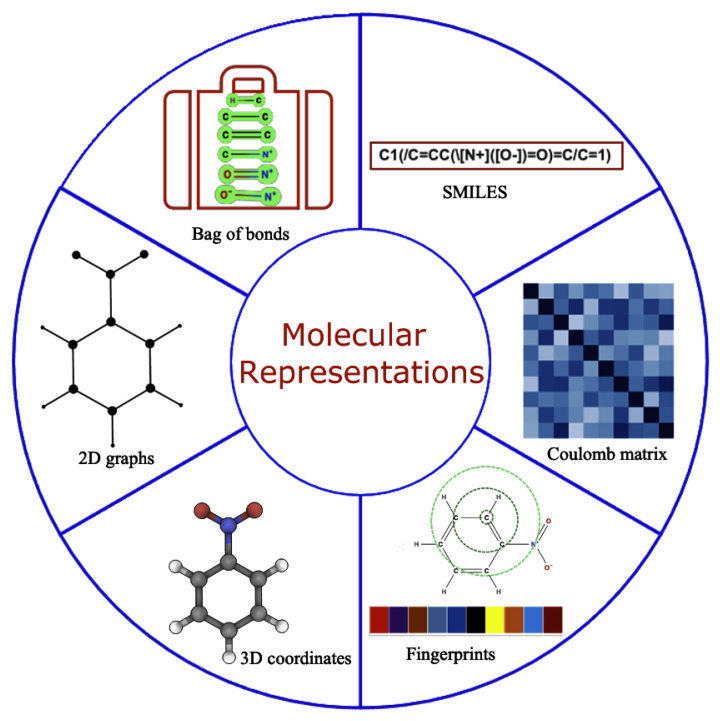
Molecular representation with all possible formulation used in the literature for predictive and generative modeling.

**Figure 3 molecules-26-06761-f003:**
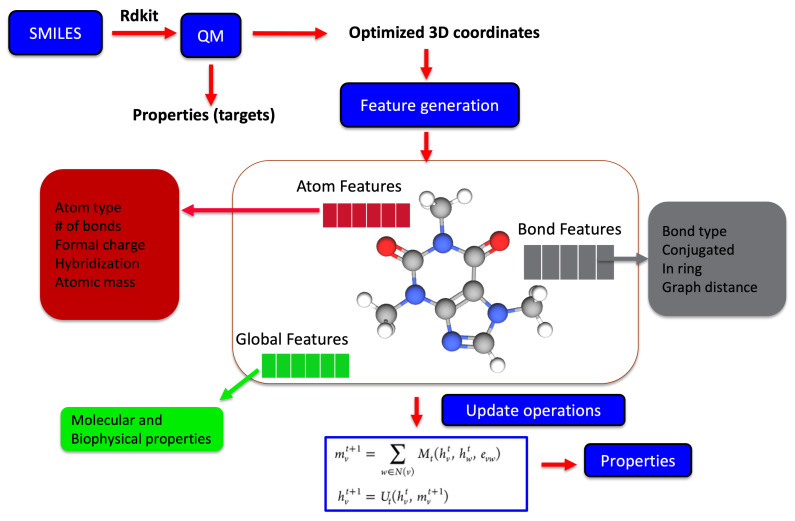
The iterative update process used for learning a robust molecular representation either based on 2D SMILES or 3D optimized geometrical coordinates from physics-based simulations. The molecular graph is usually represented by features at the atomic level, bond level, and global state, which represents the key properties. Each of these features are iteratively updated during the representation learning phase, which are subsequently used for the predictive part of model.

**Figure 4 molecules-26-06761-f004:**
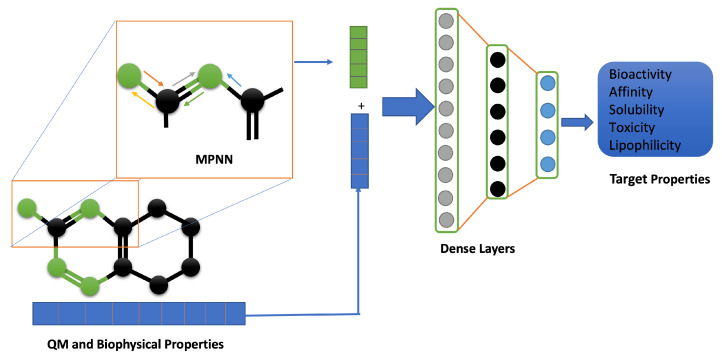
Physics-informed ML framework for predictive modeling. It takes into account the properties obtained from quantum mechanics-based simulation or from experimental data to ultimately generate features in addition to the standard process used in benchmark models (e.g., message passing neural network (MPNN).

**Figure 5 molecules-26-06761-f005:**
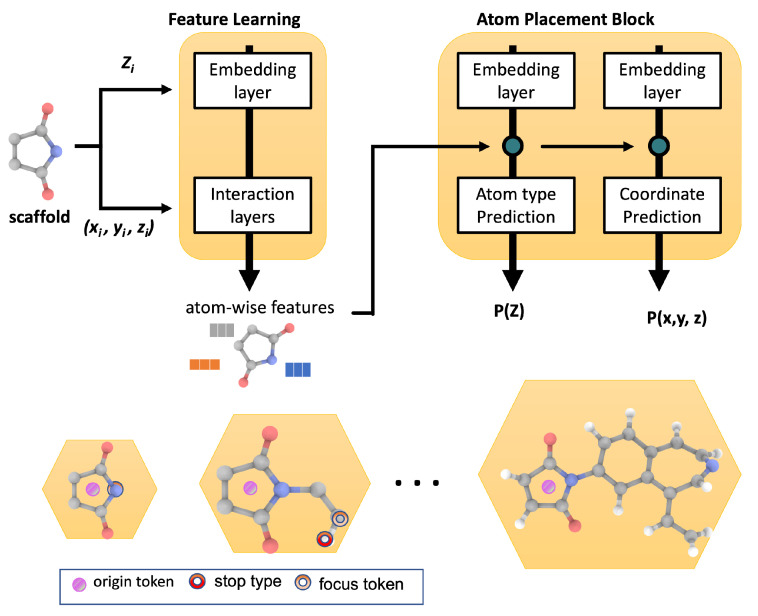
Generative model such as 3D-scaffold [69] can be used to inverse design novel candidates with desired target properties starting from core scaffold or functional group.

**Figure 6 molecules-26-06761-f006:**
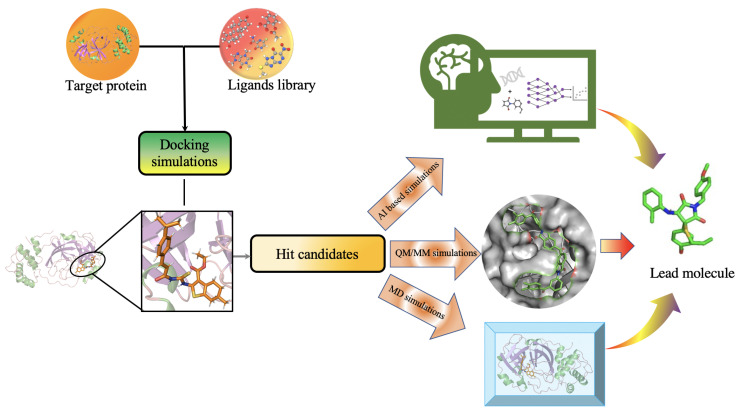
Molecular modeling methods used to study protein–ligand interactions including molecular docking simulations, molecular mechanics methods, hybrid Quantum Mechanics/Molecular Mechanics simulations, and deep learning models for the activity and affinity prediction.

**Table 1 molecules-26-06761-t001:** Highlights and benchmark of predictive ML methods, their comparison, including their key features, advantages, and disadvantages.

Methods	Key Feature	Advantage	Drawbacks
MPNN [60]	Message exchanged between the atoms depends only on the feature of the sending atom and the corresponding edge features and is independent of the representation of the atom receiving the message Generate global representation of the molecule Predicted property of the molecule is the function of global representations of the molecule Generate messages centered on the atoms	Achieved chemical accuracy in 11 out of 13 properties in QM9 data Performs well for intensive properties	Including the state of the message-receiving atom (dubbed as pair message) increases the property prediction error The message passed from atom A to atom B can be transmitted back to atom B, resulting in noise
d-MPNN [61]	Learns molecular representation centered on bonds instead of atoms Update on MPNN that combines the learned representation with the prior known fixed atomic, bond, and global molecular descriptors	Avoid noise resulting from the message being passed along any path by using directed messages Use only SMILES string to generate input representation	Does not use spatial information as a part of input features
SchNet [58]	Learns the atomistic representations of the molecules The total property of the molecule is the sum over the atomic contributions Learns representations only by using the atomic number and geometry as atom and bond features, respectively	Improves the performance on 8 out of 13 properties in QM9 data compared to MPNN Performs relatively well compared to MPNN for extensive properties Requires only the nuclear charge and nuclear coordinates for learning input representations	Relatively poor performance for intensive properties compared to MPNN Use optimized 3D coordinates
MEGNet [34]	Learns the global representations of the molecules Uses several atomic and bond properties of the atom and bond as atom and bond features Adds the global state attribute of molecule in addition to atom and bond feature	Improves the performance on all the extensive properties compared to MPNN and SchNet Works equally well for molecules and solid Provides good accuracy with RDkit-generated 3D coordinates	Larger error for intensive properties compared to MPNN It calculates MAE errors for atomization energies of U0, U, H, and G and compares with MAE on U0, U, H, and G of SchNet
SchNet-edge [80]	Edge feature also depends on the features of the atom receiving the message	Improves the accuracy of the model over SchNet/MPNN in all the properties in the QM9 dataset	Requires optimized 3D coordinates

**Table 2 molecules-26-06761-t002:** Mean absolute errors obtained from several benchmark methods on 12 different properties using the QM9 molecular dataset. Bold represents the lowest mean absolute errors among the models. * represents the property trained for respective atomization energies. Target corresponds to the chemical accuracy for each property desired from the predictive ML models.

Property	Units	MPNN	SchNet-Edge	SchNet	MegNet	Target
HOMO	eV	0.043	**0.037**	0.041	0.038 ± 0.001	0.043
LUMO	eV	0.037	**0.031**	0.034	**0.031 ± 0.000**	0.043
band gap	eV	0.069	**0.058**	0.063	0.061 ± 0.001	0.043
ZPVE	meV	1.500	1.490	1.700	**1.400 ± 0.060**	1.200
dipole moment	Debye	0.030	**0.029**	0.033	0.040 ± 0.001	0.100
polarizability	Bohr${}^{2}$	0.092	**0.077**	0.235	0.083 ± 0.001	0.100
R${}^{2}$	Bohr${}^{2}$	0.180	**0.072**	0.073	0.265 ± 0.001	1.200
U${}_{0}$	eV	0.019	0.011 *	0.014	**0.009 ± 0.000** *	0.043
U	eV	0.019	0.016 *	0.019	**0.010 ± 0.000** *	0.043
H	eV	0.017	0.011 *	0.014	**0.010 ± 0.000** *	0.043
G	eV	0.019	0.012 *	0.014	**0.010 ± 0.000** *	0.043
C${}_{v}$	cal (mol K)${}^{-1}$	0.040	0.032	0.033	**0.030 ± 0.000**	0.050

## Data Availability

Not applicable.

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
