# Peer review of "Artificial Intelligence for Autonomous Molecular Design: A Perspective"

_molecules, 2021, doi:10.3390/molecules26226761_

Round 1
Reviewer 1 Report
This review article is "okay". It is fairly well-written, but the text requires better organization of the topics. It reads like a long summary of important research articles in the field. The results and ideas from the different papers are not well-integrated or probed at a deeper level. I suggest the authors rewrite the paper with more focus on deeper coverage of the most significant results and highlighting controversies/problems in the field. More figures would also help improve the educational outreach of the paper. The authors should try to make their review paper stand out from the many other excellent review papers already published on this topic.
There are many errors in the citation of references. For example, several of the references cite preprints that have now been published as full articles (ref 38). Some references are cited incorrectly with regard to author names (ref. 47).
Author Response
We thank the reviewer for reading the manuscript and thoughtful suggestions on the organization of the paper. In the revised manuscript, we have reorganized the sections/subsections by providing additional details on the topic areas and referencing with more figures. We’ve also corrected the references in the revised version.
Reviewer 2 Report
This manuscript did not strike me as worthy of publication. Everything that is written there is quite obvious and well-known things in the chemoinformatics community.
Author Response
Response: We respectfully, disagree with the Reviewer’s comment. Such a statement without any solid base is not really helpful as the artificial intelligence-based molecular design is still not fully automated to realize its application in drug discovery or in any other applications. Our molecular design review serves as a guide for medicinal, computational chemistry and biology, software engineering, and the ML community to practice autonomous molecular design in precision medicine and drug discovery.
Round 2
Reviewer 1 Report
I appreciate the efforts of the authors to improve their manuscript. There are still many errors in the references. Please check carefully. For example, there are errors in refs 48, 50, 51, 82, 96, 99, 100, and probably many more.
Author Response
Thank you very much for the suggestions. We have checked the document carefully and corrected the error in the references.
Reviewer 2 Report
Yes, perhaps the authors are right - let's publish this manuscript.
Author Response
Thank you so much for recommending this in the current version.